# Nerve recovery from treatment with a vascularized nerve graft compared to an autologous non-vascularized nerve graft in animal models: A systematic review and meta-analysis

Berend O. Broeren[1]*, Liron S. Duraku[2], Caroline A. Hundepool[3], Erik T. Walbeehm[1], J. Michiel Zuidam[3], Carlijn R. Hooijmans[4,5], Tim De Jong[1]

1 Department of Plastic & Reconstructive Surgery, Radboud University Medical Centre, Nijmegen, The Netherlands, 2 Department of Plastic, Reconstructive and Hand Surgery, Amsterdam UMC, Amsterdam, The Netherlands, 3 Department of Plastic & Reconstructive Surgery, Erasmus MC, Rotterdam, The Netherlands, 4 Department for Health Evidence Unit SYRCLE, Radboud University Medical Centre, Nijmegen, The Netherlands, 5 Department of Anesthesiology, Radboud University Medical Centre, Nijmegen, The Netherlands

* berend.broeren@radboudumc.nl

## Abstract

### Background

Treatment of nerve injuries proves to be a worldwide clinical challenge. Vascularized nerve grafts are suggested to be a promising alternative for bridging a nerve gap to the current gold standard, an autologous non-vascularized nerve graft. However, there is no adequate clinical evidence for the beneficial effect of vascularized nerve grafts and they are still disputed in clinical practice.

### Objective

To systematically review whether vascularized nerve grafts give a superior nerve recovery compared to non-vascularized nerve autografts regarding histological and electrophysiological outcomes in animal models.

### Material and methods

PubMed and Embase were systematically searched. The inclusion criteria were as follows: 1) the study was an original full paper which presented unique data; 2) a clear comparison between a vascularized and a non-vascularized autologous nerve transfer was made; 3) the population study were animals of all genders and ages. A standardized mean difference and 95% confidence intervals for each comparison was calculated to estimate the overall effect. Subgroup analyses were conducted on graft length, species and time frames.

**Data Availability Statement:** All relevant data are within the paper and its Supporting information files.

**Funding:** The authors will receive an award from ZonMw upon publication. Partially to make open access publication possible. ZonMw is an independent institute which has no benefits from the publicated data of this article. Therefore, the funders had no role in study design, data collection and analysis, decision to publish, or preparation of the manuscript. Initials receiving author: T. De Jong Grant number: 114024159 Name funder: ZonMw URL:https://www.zonmw.nl/nl/ The funders had no role in study design, data collection and analysis, decision to publish, or preparation of the manuscript.

**Competing interests:** The authors have declared that no competing interests exist.

## Results

Fourteen articles were included in this review and all were included in the meta-analyses. A vascularized nerve graft resulted in a significantly larger diameter, higher nerve conduction velocity and axonal count compared to an autologous non-vascularized nerve graft. However, during sensitivity analysis the effect on axonal count disappeared. No significant difference was observed in muscle weight.

## Conclusion

Treating a nerve gap with a vascularized graft results in superior nerve recovery compared to non-vascularized nerve autografts in terms of axon count, diameter and nerve conduction velocity. No difference in muscle weight was seen. However, this conclusion needs to be taken with some caution due to the inherent limitations of this meta-analysis. We recommend future studies to be performed under conditions more closely resembling human circumstances and to use long nerve defects.

## Introduction

Treatment of nerve injuries proves to be a worldwide clinical challenge. Even though adequately treated, affected patients may suffer from chronic pain or lasting motor and sensory deficits [1]. For clinical situations in which it is necessary to bridge a nerve gap and when a tensionless coaptation is not possible, the current gold standard is an autologous non-vascularized (conventional) nerve graft. Treatment with a nerve graft always has a worse nerve recovery compared to primary coaptation, due to two anastomosis sides which increases the surface that needs to regenerate, ischemia of the graft and frequently a poor wound bed [2].

To improve the outcome after nerve repair with conventional nerve autografts the blood supply can be taken along with the nerve graft, the so-called vascularized nerve graft. Where a complete neurovascular bundle, which includes a nerve, its artery and its venae comitantes is harvested. The graft is placed in the middle of the nerve gap and the transected ends are sutured to the graft ends. Grafted nerves need considerable energy to regenerate and to maintain function. This energy is delivered by the intraneural vascular system, which is connected to extrinsic vessels. A vascularized graft is believed to restore these extrinsic neural blood vessels. Other theoretical reasons are: 1) the maintenance of vascularization promotes clearance of myelin debris, which results in faster remyelination; 2) the reduction of intraneural fibrosis as a result of ischemia eases axonal regeneration; 3) faster reinnervation reduces muscle atrophy [3–5].

There is no adequate clinical evidence of beneficial effect of vascularized nerve grafts except several case reports and case series [6–12]. These reported a superior two-point discrimination, sharp-blunt discrimination and warm-cold discrimination. The use of a vascularized nerve graft was first reported in 1976 by Taylor and Ham. They used 24 cm of the superficial radial nerve attached to the radial artery to reconstruct a median nerve defect [13].

Since the first publication by Taylor and Ham, many experimental studies in animal models have been reported. Vascularized nerve grafts have been successfully attempted in rats, rabbits, dogs, and other species to develop a model that is feasible, straightforward, reliable, and reproducible [14].

Nowadays, the use of vascularized nerve grafts is still debated in clinical practice because of several reasons: 1) the concern of a more significant donor site morbidity compared to conventional nerve autografts; 2) the lack of clinical evidence indicating the superiority of a vascularized nerve graft; 3) the difficulty to set up a controlled trial, due to the high heterogeneity of patients as well as nerve defects. For these reasons vascularized nerve grafts are no part of current clinical guidelines. Most institutions, including ours, use autologous nerve (cable) grafts as first option for long peripheral nerve gaps. If not enough autologous nerve is present, processed human allografts are used as second best [15–17]. However, both autologous and allografts lack vascularity.

Therefore, a systematic review and meta-analysis of animal models was conducted to investigate whether vascularized nerve grafts show a superior nerve recovery compared to non-vascularized nerve autografts regarding histological and electrophysiological factors.

## Material and methods

### Research protocol

This systematic review protocol was defined in advance and registered in an international database (PROSPERO, registration number CRD42020184363).

### Search strategy

A systematic search has been performed in the PubMed (Medline) and Embase (OVID) databases to identify all original articles. The search included studies up to 26th of May 2020. Search terms included 'nerve transfer', 'nerve graft', 'vascularized' and 'vascularization' and their synonyms in abstract and title fields (for the complete search strategy, see S1 Table). The SYRCLE search filters to identify all animal studies were used [18, 19]. Duplicates were taken out using Endnote (Clarivate Analytics, Pennsylvania, USA). Two authors (BOB and TDJ) independently screened all titles and abstracts for their relevance utilizing predetermined inclusion and exclusion criteria. A reference- and citation check of the remaining studies was conducted manually to acquire potentially missed relevant articles. Afterward, the full text of the relevant articles was screened for final selection. Contradictory judgments were resolved by consensus discussion. No language or date restrictions were applied.

### Inclusion and exclusion criteria

Articles were included when 1) the study was an original full paper which presented unique data; 2) a clear comparison between a vascularized and a non-vascularized autologous nerve transfer was made; 3) the population study were animals (all species) of all genders and ages; 4) the study investigated the effects of vascularized nerve grafts on: axonal count, diameter, nerve conduction velocity and muscle weight. No language or publication date restrictions were applied.

### Critical appraisal

All included studies were appraised using the SYRCLE's tool for assessing the risk of bias for animal studies [20]. This appraisal was done by two authors (BOB and TDJ) independently and subsequently merged by consensus. All criteria were scored a "yes" indicating a low risk of bias or a "no" indicating high risk of bias or a "?" indicating an unknown risk of bias. Baseline characteristics were: weight, age and race. Selective outcome reporting was determined by establishing if all outcome measures mentioned in Material and methods were reported in the Results section as well. To compensate for judging a lot of items as "unclear risk of bias" due to

highly inadequate reporting of experimental details on animals, methods and materials, we included two items. The first item was reporting on any measure of randomization and the second item was reporting on any measure of blinding. Here a "yes" signifies reported and a "no" means unreported.

### Data extraction

Data were in duplicate extracted from the selected studies by two authors (BOB and TDJ). The descriptive data included: publication year, first author's name, studied species, gender, total number of animals, total grafts, studied nerve, studied muscle, graft length and time points. For the meta-analysis, the mean, sd and n of the following outcomes were extracted for axonal count, diameter, nerve conduction velocity and muscle weight. When measurements of multiple locations per nerve were reported, the most distal segment of the graft was used. In case the SEM was reported it was converted to SD (SD = SEM x $\sqrt{n}$). When outcome measure data was missing, authors were contacted for additional information. When data were displayed only graphically, we used Universal Desktop Ruler software (https://avpsoft.com/products/udruler/), to determine an adequate estimation of the outcome measurements. The mean of two independent measurements was used.

### Statistical analysis

Data were analyzed using Review Manager, Version 5.4. Copenhagen: The Nordic Cochrane Centre, The Cochrane Collaboration. Meta-analysis was performed for all four outcome measurements by calculating the standardized mean difference (SMD) between vascularized and conventional grafts. Whenever a comparison reported an SD of 0 it was excluded from meta-analysis. A random effects model was applied, taking into account the accuracy of independent studies and the variation among studies and weighing all studies accordingly. Heterogeneity was measured using $I^2$. Subgroup analyses were performed for different species (rabbit and rat), different graft length (0–2 cm, 2–4 cm and 4 > cm) and different time frames (0–2 months, 2–4 months and 4 > months). The results of subgroup analysis were only interpreted when groups consisted of 3 or more individual studies.

Funnel plots, egger regression and Trim and Fill analysis were used to search for evidence for publication bias if at least 10 or more studies per outcome. Because SMDs may cause funnel plot distortion, we plotted the SMD against a sample size-based precision estimate($1/\sqrt{(n)}$).

To assess the robustness of our findings, a sensitivity analysis was performed. We evaluated the impact of excluding studies which used animals as their own control group.

## Results

### Study selection process

The search strategy presented in S1 Table retrieved 303 records, including 131 in PubMed and 172 in Embase. After removing duplicates, 203 articles appeared to be unique (Fig 1 shows a consort flow chart). After title abstract screening, 28 studies entered the full text screening phase. Finally, 14 articles were included in the review.

### Study quality and risk of bias

This review clearly revealed that methodological details of animal experiments were often poorly reported. Reporting about any randomization and blinding measures taken in the conducted studies was respectively 21% (3 out of 14 publications).

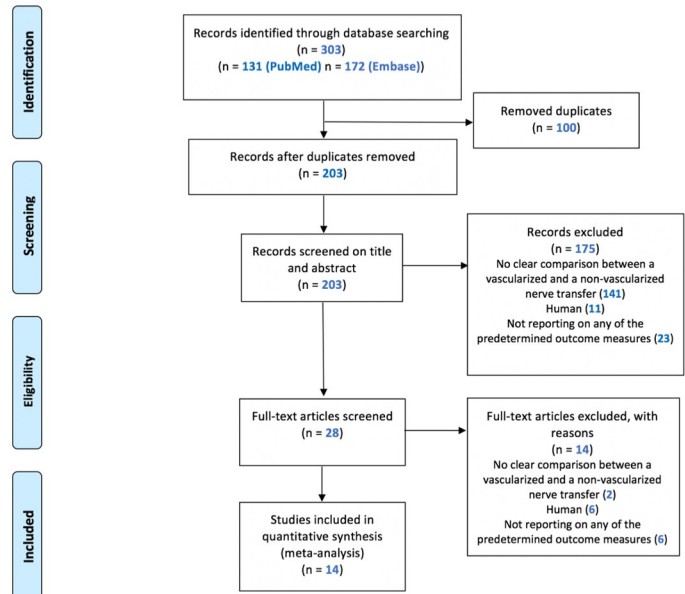

**Fig 1. Flow chart of the study selection.**

The general results of our risk of bias assessment of the included references in this review are presented in Fig 2. Poor reporting of essential methodological details in most animal experiments resulted in an unclear risk of bias in the majority of studies. Risk of bias was scored separately for the 3 studies that used animals as their own control group because some aspects were not applicable (Fig 3).

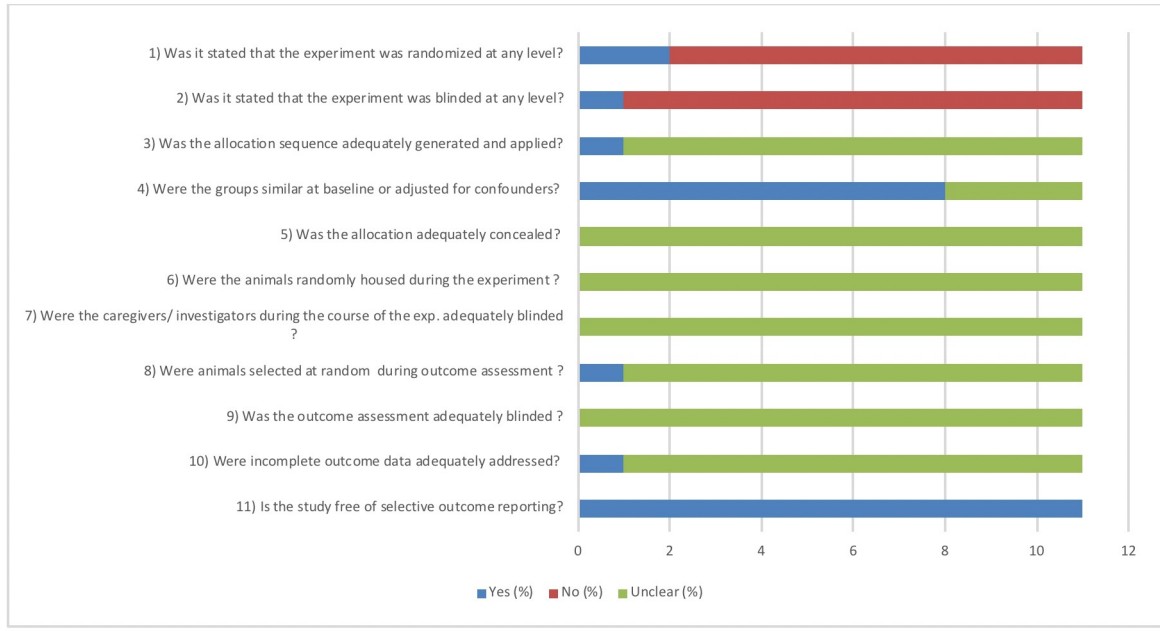

**Fig 2. Results of the risk of bias assessment of 11 included studies in this systematic review.** The first two items assess study quality by scoring reporting, a "yes" score indicates reported and a "no" score indicates unreported. The other items assessed risk of bias, with "yes" indicating low risk of bias, "no" high risk of bias, and "?" unclear risk of bias.

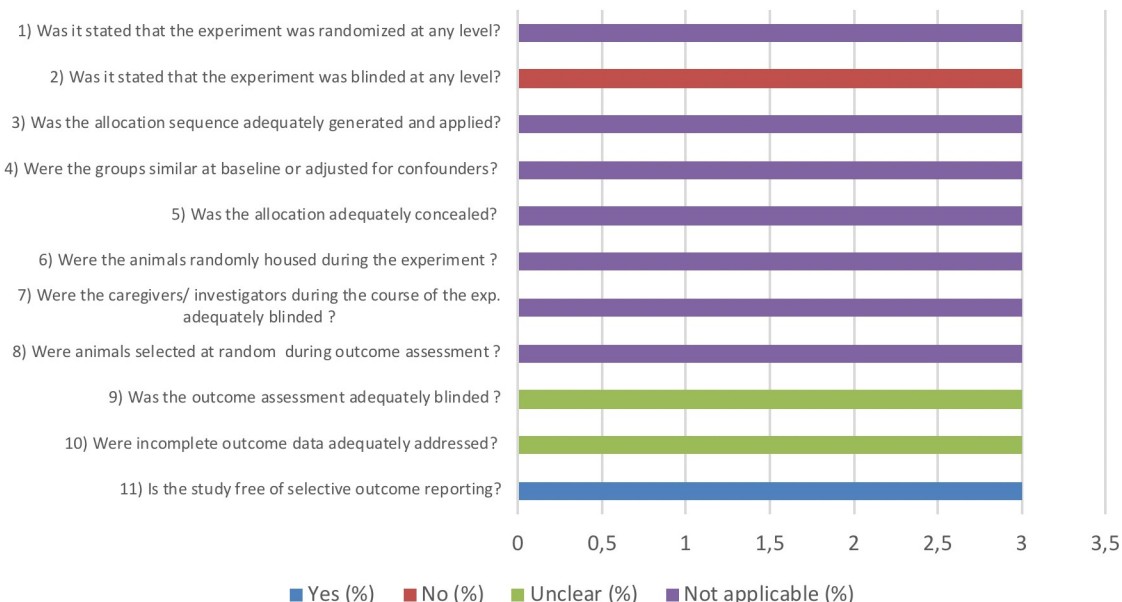

**Fig 3. Results of the risk of bias assessment of the 3 included studies in this systematic review where animals were their own control group.** The first two items assess study quality by scoring reporting, a "yes" score indicates reported and a "no" score indicates unreported. The other items assessed risk of bias, with "yes" indicating low risk of bias, "no" high risk of bias, and "?" unclear risk of bias.

## Study characteristics

The characteristics of the 14 included publications are shown in Table 1 [21–34]. All studies used either a rabbit (57%) or rat (43%) model. Notably, more than half the studies did not report gender (8 out of 14 studies). Out of the remaining studies 3 used females, 2 used males and in one both females and males were used. The sciatic nerve was the most commonly used nerve (50%), followed by the median nerve (29%), facial nerve (7%), peroneal nerve (7%) and auricular nerve (7%).

## Axonal count

Data on axonal count could be retrieved from 11 independent studies containing 37 comparisons [22–27, 29–33]. Seven comparisons had to be excluded because not all outcome data was available, for example the SD. Out of the remaining 30 experiments conducted, data obtained from rabbits and rats was both 50%. In total 352 grafts were placed in 309 animals.

There was a variation in graft length from 7 to 60 mm. The graft length was unreported in two of the comparisons. Data were extracted at different time points varying between 21 and 250 days. The way axon count was measured differed. 27 out of the 30 comparisons used a field of interest of the nerve to determine the mean axonal count compared to three using the complete diameter. Generally, it was not mentioned how this field was chosen. Next to that, different formulas and computer programs were used to calculate the number of axons, such as a VAX or Compaq computer.

Overall analysis showed a significant difference in favor of treatment with a vascularized nerve graft (SMD, 0.46 [95% CI 0.06 to 0.86], N = 30) (Fig 4). The overall between study heterogeneity was moderate to high at $I^2 = 61\%$.

Subgroup analyses revealed no differences in graft length, species and time frames when comparing axonal count between vascularized and conventional nerve autografts. The graft length middle group consisted of too few studies for subgroup analyses. (S1–S3 Figs).

**Table 1. The characteristics of all 14 included references.**

| Reference | Outcome measurements | Species | Gender | Animals | Grafts | Nerve | Muscle | Graft size (mm) | Time points (days) |
|---|---|---|---|---|---|---|---|---|---|
| Bertelli et al., 1996 | Muscle weight | Rat | Female | 70 | 70 | Median | FCR | 20 | 95, 120, 150, 210, 360 |
| Donzelli et al., 2016 | Axonal count | Rabbit | Male | 20 | 20 | Sciatic | | | 30, 90 |
| | Diameter | | | | | | | | |
| Hems et al., 1992 | Axonal count | Rabbit | NR | 8 | 8 | Peroneal | | 50 | 250 |
| Kanaya et al., 1992 | Axonal count | Rat | Female | 22 | 22 | Sciatic | Tibialis anterior | 25 | 84 |
| | Nerve conduction velocity | | | | | | | | |
| | Muscle weight | | | | | | | | |
| Kawai et al., 1990 | Axonal count | Rabbit | NR | 34 | 67 | Median | | 20,40, 60 | 56, 168 |
| | Diameter | | | | | | | | |
| Koshima et al., 1985 | Axonal count | Rat | Male | 38 | 38 | Sciatic | | 15 | 28, 56, 84, 112, 140, 168 |
| | Diameter | | | | | | | | |
| Koshima.2 et al., 1985 | Axonal count | Rat | NR | 74 | 74 | Sciatic | | 15 | 21, 28, 35, 42, 49, 56, 84,112, 140, 168, 224 |
| | Diameter | | | | | | | | |
| Mani et al., 1992 | Diameter | Rabbit | Male/ Female | 11 | 11 | Sciatic | | 30 | 308 |
| Matsumine et al., 2013 | Axonal count | Rat | NR | 14 | 14 | Median | | 7 | 210 |
| | Diameter | | | | | | | | |
| Ozcan et al., 1993 | Axonal count | Rabbit | Female | 10 | 10 | facial | | 10 | 84 |
| | Diameter | | | | | | | | |
| Seckel et al., 1986 | Axonal count | Rat | NR | 13 | 26 | Sciatic | | 10 | 21, 28, 42 |
| Shibata et al., 1988 | Axonal count | Rabbit | NR | 39 | 39 | Median | | 30 | 70, 168 |
| | Diameter | | | | | | | | |
| | Nerve conduction velocity | | | | | | | | |
| Tark et al., 2001 | Axonal count | Rabbit | NR | 33 | 66 | Sciatic | | 40 | 56, 84, 112 |
| Zhu et al., 2015 | Nerve conduction velocity | Rabbit | NR | 6 | 6 | Auricular | | 20 | 112 |

NR: not reported.

FCR: flexor carpi radialis.

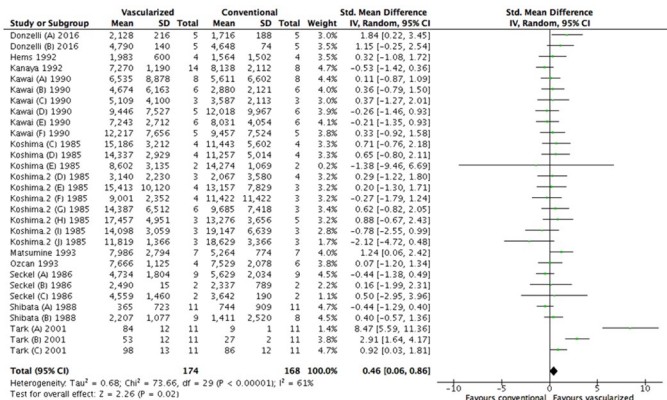

**Fig 4. Forest plot of the effect of treatment with a vascularized nerve graft on axonal count.** Data are presented as standardized mean difference (SMD) and 95% confidence intervals (CL).

## Diameter

Eight studies, containing 31 comparisons, reported nerve fiber diameter on histological examination [22, 25–30, 32]. Since not all data was available, 10 of the 31 comparisons had to be excluded. Rabbits and rats were used in 52% and 48% respectively. All 21 comparisons combined, a total of 148 animals were operated on, resulting in 185 grafts that met our selection criteria. Graft length varied between 7 and 60 mm. In one of the studies, it was unclear which graft length was used. The time points at which data were extracted ranged from 21 to 308 days.

Analysis of all 21 included comparisons showed a significantly larger diameter after treatment with a vascularized nerve graft (SMD, 0.59 [95% CI 0.16 to 1.02], N = 21) (Fig 5). Study heterogeneity was moderate ($I^2 = 36\%$).

The subgroup analysis for graft length could not be interpreted because both the middle and long group consisted of fewer than 3 studies.

For species however, there was a significant difference in diameter comparing rabbits and rats showing a more positive result in rats (SMD 0.13 [95% CI -0.28 to 0.54], N = 11; $I^2 = 11\%$ compared to SMD 1.40 [95% CI 0.74 to 2.06], N = 10; $I^2 = 3\%$; P = 0.005). Rats showed a significant larger nerve fiber diameter in vascularized grafts compared to conventional grafts (S4 Fig).

A significant difference in diameter could not be found comparing different time frames (S5 Fig).

## Nerve conduction velocity

Data on nerve conduction velocity could be extracted from 3 studies containing 4 comparisons [24, 32, 34]. Three comparisons used a rabbit model. A total of 74 animals were operated on, resulting in 74 grafts that met our selection criteria. Graft length ranged from 20 to 30 mm. Outcomes were measured at time points between 70 and 168 days.

Overall, analysis showed treatment with a vascularized nerve graft resulted in a significantly higher nerve conduction velocity (SMD, 1.19 [95% CI 0.19 to 2.19], N = 4) (Fig 6). Between

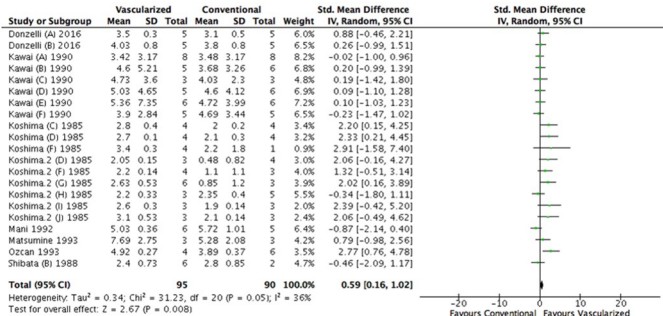

**Fig 5. Forest plot of the effect of treatment with a vascularized nerve graft on diameter.** Data are presented as standardized mean difference (SMD) and 95% confidence intervals (CL).

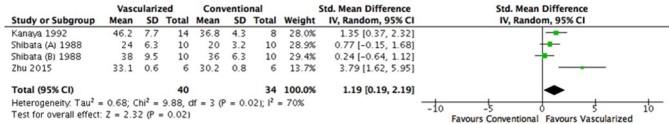

**Fig 6. Forest plot of the effect of treatment with a vascularized nerve graft on nerve conduction velocity.** Data are presented as standardized mean difference (SMD) and 95% confidence intervals (CL).

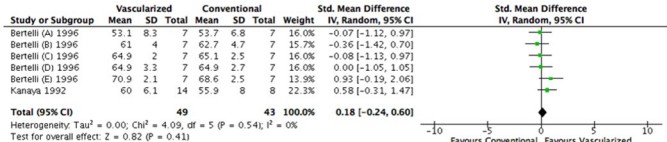

**Fig 7. Forest plot of the effect of treatment with a vascularized nerve graft on muscle weight.** Data are presented as standardized mean difference (SMD) and 95% confidence intervals (CL).

studies, heterogeneity was high ($I^2$ = 79%). There were not enough studies to perform a subgroup analysis.

## Muscle weight

Two studies, containing 6 comparisons, assessed muscle weight [21, 24]. A total of 92 animals, all rats, were operated on, resulting in 92 grafts. The two graft lengths used were 20 and 25mm. The varying time points at which data were extracted were between 84 and 360 days.

Overall, no significant difference was found between the treatment groups (SMD, 0.18 [95% CI -0,24 to 0,60], N = 6), $I^2$ was 0% (Fig 7). There were not enough studies to perform a subgroup analysis.

## Sensitivity analyses

**Axonal count.** Exclusion of the studies in which animals were their own control group altered our results significantly. The previous effect in favor of a vascularized nerve graft compared to a conventional nerve autograft was no longer available (SMD 0.26 [95% CL -0.09 to 0.62], N = 18), heterogeneity was $I^2$ = 17% (Fig 8). Conclusions of all subgroup analyses appeared to be robust (S6–S8 Figs).

**Diameter.** Exclusion of the studies in which animals were their own control group did not alter our results significantly. A significant difference in favor of a vascularized nerve graft compared to a conventional nerve autograft was found (SMD 1.03 [95% CL 0.39 to 1.68], N = 15), heterogeneity was $I^2$ = 46% (Fig 9).

Next to that, the result of the subgroup analysis on species was altered. No significant difference in favor of rats was found. (SEM 0.39 [95% CI -0.68 to 1.45], N = 5; $I^2$ = 62% compared to SEM 1.40 [95% CI 0.74 to 2.06], N = 10; $I^2$ = 3%; P = 0.13) (S9 Fig). Other conclusions appeared to be robust (S10 Fig).

## Publication bias analysis

Publication bias was assessed for axonal count only, because all other outcomes contained fewer than 10 studies.

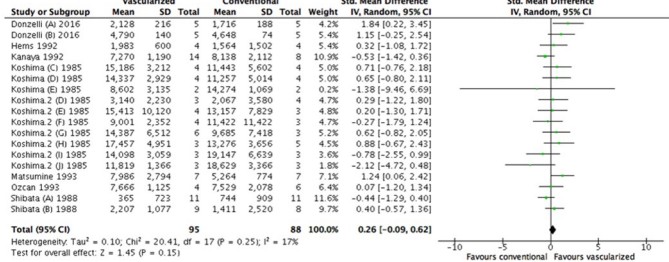

**Fig 8. Sensitivity analysis, forest plot of the effect of treatment with a vascularized nerve graft on axonal count.** Data are presented as standardized mean difference (SMD) and 95% confidence intervals (CL).

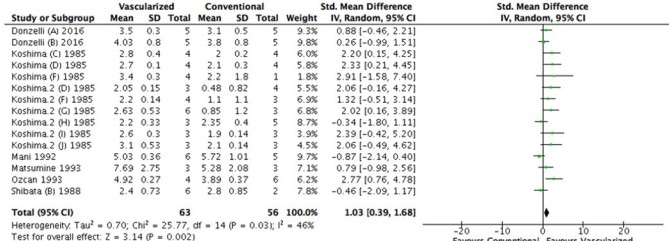

**Fig 9. Sensitivity analysis, forest plot of the effect of treatment with a vascularized nerve graft on diameter.** Data are presented as standardized mean difference (SMD) and 95% confidence intervals (CL).

**Axonaffl count.** The funnel plot suggested some asymmetry. Duval and Tweedie's Trim and Fill analysis resulted in 6 extra data points (see Fig 10), indicating the presence of publication bias and some overestimation of the identified summary effect size.

## Discussion

This review suggests that a vascularized nerve graft does result in a significantly better nerve recovery compared to non-vascularized nerve autografts in animal models regarding the outcome measurements nerve fiber diameter, nerve conduction velocity and axonal count. However, the effect on axonal count did not appear to be very robust as after sensitivity analysis the effect was no longer present. Muscle weight did not differ between vascularized and non-vascularized grafts. Subgroup analysis indicated that the effect of vascularized graft on nerve fiber diameter is larger in rats compared to rabbits. However, this difference disappeared after sensitivity analysis.

A superior axon count and diameter was to be expected. A vascularized graft is believed to reduce intraneural fibrosis secondary to ischemia, which eases axonal regeneration. We expected a superior muscle weight as well because of a faster reinnervation reducing denervation of muscle atrophy. We think the small number of studies, that reported on muscle weight and therefore could be used for meta-analysis, is the reason for a lacking effect.

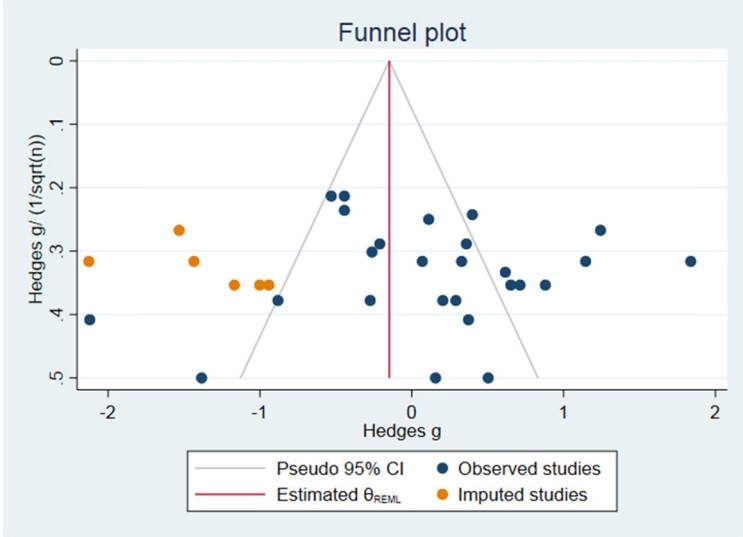

**Fig 10. Publication bias.**

There is a lot of discussion on what the best outcome measurement for nerve regeneration is. Until this day there is no proper "gold standard" to test nerve recovery, although the ultimate goal of nerve recovery is to maximize sensation and motion. The most commonly used outcome measurement for sensation is the von Frey test [35]. For motion, walking track analysis was believed to be the best overall assessment [36–39]. At the moment it is rarely used and some would say it is even obsolete. Additionally, walking track analysis does not reflect maximum muscle force capacity. Others say the most precise measurement is the isometric response of muscle to tetanic contraction [40]. The authors are aware of the fact that histomorphometry, electrophysiology and axonal count in particular may have a limited correlated to the real functional recovery of sensation or motion [41]. However, despite the fact that these functional outcomes have the greatest clinical relevance, no data on functional recovery are present in the current literature.

This present meta-analysis of animal studies is, to the best of our knowledge, the first of its kind. Only some human case reports exist to try to put our findings into a broader perspective [8, 11, 42]. The clinical observations in these human case reports did not include the outcome measurements of this review. Nevertheless, all showed a superior sensory recovery in vascularized nerve grafts compared to conventional nerve grafts using different outcome measurements, such as the presence of a sharp/blunt discrimination, cold intolerance, the Tinel's sign and the Semmes-Weinstein monofilament test.

Notably, clinical case reports found that vascularized nerve grafts give a better recovery in large nerve grafts compared to conventional nerve autografts. Terzis et al. [43] showed that a vascularized nerve graft successfully bridges a nerve defect longer than 13 cm where conventional nerve grafts generally fail. Also Xu et al. [44] and Okinaga et al. [45] concluded that when the graft length was short, the results were not significantly in favor of a vascularized nerve graft. However, we did not find a difference in recovery between various graft lengths in this meta-analysis.

As previously discussed a more significant donor site morbidity is one of the concerns surrounding a vascularized nerve graft. Donor site morbidity mainly depends on the harvested nerve, and the extend of the needed dissection. Therefore, we think it should also be assessed per case if the advantages of a vascularized graft justify the disadvantages of longer operative time and an enlarged donor site.

## Limitations of this review

Firstly, our risk of bias analysis showed that most studies reported poorly on important methodological details. Therefore, most of the risk of bias items assessed had to be scored as unclear risk of bias. Even though this is quite commonly seen in animal studies, it is something to be taken into account [46]. The absence of reporting such methodological details could, to a certain extent, indicate the negligence of using these methods to minimize bias and confounding [47]. This can seriously hamper the possibility to draw reliable conclusions from the included animal studies.

Secondly, the number of studies included in this meta-analysis is relatively low, especially on nerve conduction velocity and muscle weight. This resulted in subgroups being relatively small, even to the extent that some subgroup analysis could not be interpreted. Furthermore, heterogeneity was moderate to high. However, because of their explorative nature a moderate to high heterogeneity between animal studies is expected.

To account for anticipated heterogeneity, we used a random effects model, conducted sensitivity analyses and explored the suggested causes for between study heterogeneity by means of subgroup analyses. Exploring this heterogeneity is one of the added values of meta-analyses

of animal studies and might help to inform the design of future animal studies and subsequent clinical trials.

Thirdly, the graft length used to repair a nerve defect in rat and rabbit models is presumably smaller than those needed in humans. Therefore, the results shown in these animal experiments might not be correlated with the expected clinical outcomes.

Fourthly, a possible reason for heterogeneity could be the use of animals as their own control in some studies. Therefore, a sensitivity analysis was performed. This led to 3 studies being excluded because animals were used as their own control group. When Kawai et al. [25], Seckel et al. [31] and Tark et al. [33] were excluded there was not a significant difference in axonal count in favor of vascularized nerve grafts compared to conventional nerve autografts.

Fifthly, histomorphometry is difficult to compare between different laboratories, because other methods to measure the outcome were used. To compensate for these differences, we used a standardized mean difference for our meta-analysis. Over the years methods have evolved from manually calculating axonal count from a light microscopic photograph to a computer calculated estimate. The methods used by the studies in this review vary as well. Searching the publication databases, we found little evidence on which one is the best or on a clear sensitivity or specificity for these methods. Kim et al. [48] concluded that semi-automated method for counting axons in transmission electron microscopic images were strongly correlated with those of conventional counting methods and showed excellent reproducibility. Nevertheless, the techniques for histomorphometry will always be an estimation and therefore prone to bias.

Lastly, the presence of publication bias was identified. Our funnel plot suggested some asymmetry and Duval and Tweedie's Trim and Fill analysis predicts some overestimation of the identified summary effect size of axonal count.

## Conclusion

Treating a nerve gap with a vascularized graft results in superior nerve recovery compared to non-vascularized autografts nerve grafts in three out of four outcome measurements. However, this conclusion needs to be taken with some caution due to the inherent limitations of this meta-analysis. In addition, we recommend future studies to be performed under conditions more closely resembling human circumstances and to use long nerve grafts. Furthermore, we underline that future studies should use the Gold Standard Publication Checklist or ARRIVE guidelines to improve the reporting and methodological quality of animal studies [49, 50]. This is essential to improve the quality of the evidence presented in animal studies and the successful translation to humans in a clinical setting.

## Supporting information

**S1 Fig. Subgroup analysis by graft length on axonal count.** Data are presented as standardized mean difference (SMD) and 95% confidence intervals (CL). Short group vs. long group P = 0.07.
(TIF)

**S2 Fig. Subgroup analysis by species on axonal count.** Data are presented as standardized mean difference (SMD) and 95% confidence intervals (CL). Rabbit group vs. rat group P = 0.06.
(TIF)

**S3 Fig. Subgroup analysis by time frames on axonal count.** Data are presented as standardized mean difference (SMD) and 95% confidence intervals (CL). Short group vs. middle group

P = 0.81. short group vs. long group P = 0.45.
(TIF)

**S4 Fig. Subgroup analysis by species on diameter.** Data are presented as standardized mean difference (SMD) and 95% confidence intervals (CL). Rabbit group vs. rat group P = 0.005.
(TIF)

**S5 Fig. Subgroup analysis by time frames on diameter.** Data are presented as standardized mean difference (SMD) and 95% confidence intervals (CL). Middle group vs. short group P = 0.12. Middle group vs. long group P = 0.08.
(TIF)

**S6 Fig. Sensitivity analysis subgroup analysis by graft length on axonal count.** Data are presented as standardized mean difference (SMD) and 95% confidence intervals (CL).
(TIF)

**S7 Fig. Sensitivity analysis subgroup analysis by species on axonal count.** Data are presented as standardized mean difference (SMD) and 95% confidence intervals (CL). Rabbit group vs. rat group P = 0.62.
(TIF)

**S8 Fig. Sensitivity analysis subgroup analysis by time frames on axonal count.** Data are presented as standardized mean difference (SMD) and 95% confidence intervals (CL).
(TIF)

**S9 Fig. Sensitivity analysis subgroup analysis by species on diameter.** Data are presented as standardized mean difference (SMD) and 95% confidence intervals (CL). Rabbit group vs. rat group P = 0.13.
(TIF)

**S10 Fig. Sensitivity analysis subgroup analysis by time frames on diameter.** Data are presented as standardized mean difference (SMD) and 95% confidence intervals (CL). Middle group vs. long group P = 0.24.
(TIF)

**S11 Fig. Raw data.**
(PDF)

**S1 Table. Search strategy.**
(TIFF)

**S1 Checklist.**
(DOC)

## Acknowledgments

The authors would like to thank Mrs On Ying Chan (Health Sciences reference librarian, Radboud University) for assisting with the development of the search strategy.

## Author Contributions

**Conceptualization:** Berend O. Broeren, Liron S. Duraku, Caroline A. Hundepool, Erik T. Walbeehm, J. Michiel Zuidam, Tim De Jong.

**Data curation:** Berend O. Broeren, Tim De Jong.

**Formal analysis:** Berend O. Broeren, Carlijn R. Hooijmans.

**Funding acquisition:** Tim De Jong.

**Investigation:** Berend O. Broeren.

**Methodology:** Berend O. Broeren.

**Project administration:** Berend O. Broeren.

**Supervision:** Tim De Jong.

**Validation:** Carlijn R. Hooijmans.

**Visualization:** Berend O. Broeren.

**Writing – original draft:** Berend O. Broeren.

**Writing – review & editing:** Liron S. Duraku, Caroline A. Hundepool, Erik T. Walbeehm, J. Michiel Zuidam, Carlijn R. Hooijmans, Tim De Jong.

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
