## [Decision Letter · Decision Letter 0]

23 Jun 2021

PONE-D-21-15417

Nerve recovery from treatment with a vascularized nerve graft compared to an autologous non-vascularized nerve graft in animal models: a systematic review and meta-analysis

PLOS ONE

Dear Dr. Broeren,

Thank you for submitting your manuscript to PLOS ONE. After careful consideration, we feel that it has merit but does not fully meet PLOS ONE’s publication criteria as it currently stands. Therefore, we invite you to submit a revised version of the manuscript that addresses the points raised during the review process.

We look forward to receiving your revised manuscript.

Kind regards,

Leila Harhaus

Academic Editor

PLOS ONE

Journal Requirements:

Reviewers' comments:

Reviewer's Responses to Questions

**Comments to the Author**

1. Is the manuscript technically sound, and do the data support the conclusions?

Reviewer #1: Yes

Reviewer #2: Partly

2. Has the statistical analysis been performed appropriately and rigorously? 

Reviewer #1: Yes

Reviewer #2: N/A

3. Have the authors made all data underlying the findings in their manuscript fully available?

Reviewer #1: Yes

Reviewer #2: Yes

4. Is the manuscript presented in an intelligible fashion and written in standard English?

Reviewer #1: Yes

Reviewer #2: Yes

5. Review Comments to the Author

Reviewer #1: Nerve recovery from treatment with a vascularized nerve graft compared to an autologous non-vascularized nerve graft in animal models: a systematic review and meta-analysis

This manuscript is a metanalysis comparing vascularized vs. no vascularized nerve graft. The article fits the scope of the journal. The title appropriately summarizes the article contents. The article is well-structured, the language is adequate. The references are uniform and according to the instructions for authors. The main concern is the heterogenous data with poor methodological details from the different manuscripts this paper is based on.

Please consider the next suggestions/clarification to improve the manuscript:

- Axonal account and diameter: were a uniform histological technique regarding the axonal account? Paper’s years of publication range from 1988 until 2016. Although these histological techniques are not novel, the sensitivity and specific of this technique has been improved during the last 30 years. Do the authors think it could exist a bias comparing these studies?

- I have a similar concern regarding the three studies with neurophysiology. This technique needs an experience neurologist or neurophysiologist to get reliable data.

I recommend the publication of the manuscript after the above comments are addressed. We thanks the authors the great effort needed for this study.

Reviewer #2: Thank you for your work and efforts to do a meta analysis.

What is the personal guidelines in your institution?

Please add your guideline into this paper.

The figures are in poor resolution, blurred and not clear.

Kindly ask you improve the resolution.

6. PLOS authors have the option to publish the peer review history of their article (what does this mean?). If published, this will include your full peer review and any attached files.

Reviewer #1: No

Reviewer #2: No

---

## [Author Response · Author response to Decision Letter 0]

2 Aug 2021

Rebuttal letter

PONE-D-21-15417

“Nerve recovery from treatment with a vascularized nerve graft compared to an autologous non-vascularized nerve graft in animal models: a systematic review and meta-analysis”

Berend O. Broeren1*, Liron S. Duraku2, Caroline A. Hundepool2, Erik T. Walbeehm1, J. Michiel Zuidam2, Carlijn R. Hooijmans3,4, Tim De Jong1

PLOS ONE

Leila Harhaus 

Academic Editor

PLOS ONE

Dear Ms Harhaus,

We thank you and the reviewers for a careful reading and the constructive comments regarding our manuscript and for the opportunity to revise and resubmit. We have addressed all recommendations and suggestions to further improve the manuscript. On behalf of my co-authors, I thank you for considering this revised manuscript for publication. We appreciate your time and look forward to your response.

Yours sincerely,

Berend Broeren (corresponding author)

berend.broeren@xs4all.nl

We have added our raw data using a supporting file and included a caption at the end of the manuscript. 

We have included a caption for each figure immediately following the paragraph in which the figure was first cited with a label, title and legend. 

We have included a caption for each supporting information file at the end of the manuscript with a label, title and legend. In-text citations were matched accordingly. 

Comments to the Author

1. Is the manuscript technically sound, and do the data support the conclusions?

Reviewer #1: Yes

Reviewer #2: Partly

2. Has the statistical analysis been performed appropriately and rigorously? 

Reviewer #1: Yes

Reviewer #2: N/A

3. Have the authors made all data underlying the findings in their manuscript fully available?

Reviewer #1: Yes

Reviewer #2: Yes

4. Is the manuscript presented in an intelligible fashion and written in standard English?

Reviewer #1: Yes

Reviewer #2: Yes

5. Review Comments to the Author

Reviewer #1: 

Nerve recovery from treatment with a vascularized nerve graft compared to an autologous non-vascularized nerve graft in animal models: a systematic review and meta-analysis

This manuscript is a metanalysis comparing vascularized vs. no vascularized nerve graft. The article fits the scope of the journal. The title appropriately summarizes the article contents. The article is well-structured, the language is adequate. The references are uniform and according to the instructions for authors. The main concern is the heterogenous data with poor methodological details from the different manuscripts this paper is based on.

Please consider the next suggestions/clarification to improve the manuscript:

- Axonal account and diameter: were a uniform histological technique regarding the axonal account? Paper’s years of publication range from 1988 until 2016. Although these histological techniques are not novel, the sensitivity and specific of this technique has been improved during the last 30 years. Do the authors think it could exist a bias comparing these studies?

We appreciate the positive comments of Reviewer #1 and share the concerns regarding the heterogenous data and the poor methodological details from the different manuscripts this paper is based on. Therefore, it was extensively described in the section ‘limitations of this study” and the recommendation to use the Gold Standard Publication Checklist or ARRIVE guidelines for future animal studies to improve the reporting and methodological quality of animal studies in our conclusion was made. As to the uniformity of histological technique to measure axonal count, we added a section in the “limitations of this study” section. Different techniques were used by the studies in this review. To compensate for these differences, we used a standardized mean difference for our meta-analysis. The statement made by Reviewer #1 that these techniques have evolved is true and the sensitivity and specific of this technique may have been improved during the last 30 years. Over the years methods have evolved from manually calculating axonal count from a light microscopic photograph to a computer calculated estimate. Searching the publication databases, we found little evidence on which one is the best or on a clear sensitivity or specificity for these methods. Kim et al. concluded that semi-automated method for counting axons in transmission electron microscopic images were strongly correlated with those of conventional counting methods and showed excellent reproducibility. Nevertheless, the techniques for histomorphometry will always be an estimation and therefore prone to bias.

- I have a similar concern regarding the three studies with neurophysiology. This technique needs an experience neurologist or neurophysiologist to get reliable data.

The studies where neurophysiology was one of the outcome measurements did not mention the skillset of the person who assessed the outcomes. The data could possibly be biased but this cannot be established with the information at hand. 

I recommend the publication of the manuscript after the above comments are addressed. We thanks the authors the great effort needed for this study.

Reviewer #2: 

What is the personal guidelines in your institution? Please add your guideline into this paper.

Although we don’t have a clinical guideline for peripheral nerve gaps, we have added information of the clinical practice of our institution in the introduction of the manuscript. 

The use of vascularized nerve grafts is debated in our medical centers because of the lack of clinical evidence indicating the superiority of a vascularized nerve graft, the concern of a more significant donor site morbidity compared to conventional nerve autografts and the complexity of the surgery. 

The figures are in poor resolution, blurred and not clear.

Kindly ask you improve the resolution.

All figures have been improved using the PACE digital diagnostic tool to make sure the resolution improved and the PLOS requirements were met.

---

## [Decision Letter · Decision Letter 1]

22 Oct 2021

PONE-D-21-15417R1Nerve recovery from treatment with a vascularized nerve graft compared to an autologous non-vascularized nerve graft in animal models: a systematic review and meta-analysisPLOS ONE

Dear Dr. Broeren,

Thank you for submitting your manuscript to PLOS ONE. After careful consideration, we feel that it has merit but does not fully meet PLOS ONE’s publication criteria as it currently stands. Therefore, we invite you to submit a revised version of the manuscript that addresses the points raised during the review process.

We look forward to receiving your revised manuscript.

Kind regards,

Leila Harhaus

Academic Editor

PLOS ONE

Journal Requirements:

Reviewers' comments:

Reviewer's Responses to Questions

**Comments to the Author**

1. If the authors have adequately addressed your comments raised in a previous round of review and you feel that this manuscript is now acceptable for publication, you may indicate that here to bypass the “Comments to the Author” section, enter your conflict of interest statement in the “Confidential to Editor” section, and submit your "Accept" recommendation.

Reviewer #3: (No Response)

2. Is the manuscript technically sound, and do the data support the conclusions?

Reviewer #3: Partly

3. Has the statistical analysis been performed appropriately and rigorously? 

Reviewer #3: Yes

4. Have the authors made all data underlying the findings in their manuscript fully available?

Reviewer #3: Yes

5. Is the manuscript presented in an intelligible fashion and written in standard English?

Reviewer #3: Yes

6. Review Comments to the Author

Reviewer #3: The study provided an interesting overview of the field of vascularized nerve grafts, indicating that vascularized nerve grafts can achieve superior results in terms of axon count, axon diameter, and nerve conduction velocity when compared to non-vascularized nerve grafts.

However, the manuscript needs some major and minor revision before publication.

Major Points

The introduction should include a more detailed explanation of the technique and rationale behind these techniques. The author discussed the disadvantages of the vascularized nerve graft but did not discuss its advantages over the conventional ANT.

Interestingly, the author emphasizes the critical nature of functional recovery but includes no data in the manuscript, implying that vascularized nerve grafts may have an effect on functional recovery. While we are well aware of the heterogeneity associated with assessing functional recovery following peripheral nerve injury, these issues should also be discussed in the manuscript.

This manuscript provided compelling evidence that a peripheral nerve surgeon should place a greater emphasis on the vascularized nerve graft. The author, however, did not discuss the findings under "Discussion."

What is the author's opinion about the superior axon count and diameter? Why do you see the effects even for the nerve conduction velocity and not for the muscle weight?

Justifies the advantages of an enlarged donor site the disadvantages?

Can the findings of an animal study be applied to humans?

Therefore the Discussion section should be appropriately revised.

Minor Points

Abstract

Line 54: What does this sentence mean? Three out of four measurements. In which measurements is a vascularized nerve graft superior?

Introduction:

The author should give a short overview of the technique of the vascularized nerve graft and explain its rationality further -

Line 63: When is it necessary to bridge the gap – when a tensionless coaptation is not possible – this should be added

Line 67: please rephrase the sentence – the meaning is clear – but the expression has to be improved

Line 80: what did these studies show – please elaborate on the findings of the previous research

Line 88: why do you use them and not nerve tubes? When you mean Axogen, then due to the results of the ranger study – these should be added.

Line 117: what about functional regeneration? Did the studies give no information focusing on the functional regeneration compared to a "conventional" ANT

Material and Methods:

Figure 1: Can you please clarify why you exclude the 175 manuscripts to classify the arguments. Would you please adjust the figure?

Results:

Line 205; the meaning of the sentence is not clear – please rephrase

Line 210: please add the level of significance

Line 235: why here SEM and not SMD?

Discussion:

This is a critical point in line 319 – 329. However, no data on functional recovery were provided, even with a small number of studies – despite the fact that this is the most important "test" for peripheral nerve injury recovery due to its clinical relevance.

Line 371: Because histomorphometry is crucial to this review, the author should explain how the axonal count was obtained. Counting axons using random field of interest methods and counting the entire diameter have varying degrees of validity for the data.

7. PLOS authors have the option to publish the peer review history of their article (what does this mean?). If published, this will include your full peer review and any attached files.

Reviewer #3: No

---

## [Author Response · Author response to Decision Letter 1]

12 Nov 2021

Comments to the Author

1. If the authors have adequately addressed your comments raised in a previous round of review and you feel that this manuscript is now acceptable for publication, you may indicate that here to bypass the “Comments to the Author” section, enter your conflict of interest statement in the “Confidential to Editor” section, and submit your "Accept" recommendation.

Reviewer #3: (No Response)

2. Is the manuscript technically sound, and do the data support the conclusions?

Reviewer #3: Partly

3. Has the statistical analysis been performed appropriately and rigorously? 

Reviewer #3: Yes

4. Have the authors made all data underlying the findings in their manuscript fully available?

Reviewer #3: Yes

5. Is the manuscript presented in an intelligible fashion and written in standard English?

Reviewer #3: Yes

6. Review Comments to the Author

Reviewer #3: The study provided an interesting overview of the field of vascularized nerve grafts, indicating that vascularized nerve grafts can achieve superior results in terms of axon count, axon diameter, and nerve conduction velocity when compared to non-vascularized nerve grafts.

However, the manuscript needs some major and minor revision before publication.

Major Points

The introduction should include a more detailed explanation of the technique and rationale behind these techniques. The author discussed the disadvantages of the vascularized nerve graft but did not discuss its advantages over the conventional ANT.

We have added information on the technique and rationale behind the technique in the introduction of the manuscript. Page 4 line 72-74 and 76-80 

Interestingly, the author emphasizes the critical nature of functional recovery but includes no data in the manuscript, implying that vascularized nerve grafts may have an effect on functional recovery. While we are well aware of the heterogeneity associated with assessing functional recovery following peripheral nerve injury, these issues should also be discussed in the manuscript.

We have added information on the rationale behind the choice of our outcome measurements and discuss the relevance of functional outcomes. Page 16 line 350, 351

This manuscript provided compelling evidence that a peripheral nerve surgeon should place a greater emphasis on the vascularized nerve graft. The author, however, did not discuss the findings under "Discussion."

The findings were indeed discussed in the “Discussion”. 

This review suggests that a vascularized nerve graft does result in a significantly better nerve recovery compared to non-vascularized nerve autografts in animal models regarding the outcome measurements nerve fiber diameter, nerve conduction velocity and axonal count. Page 15 line 326-328

Treating a nerve gap with a vascularized graft results in superior nerve recovery compared to non-vascularized autografts nerve grafts in three out of four outcome measurements. Page 19 line 414, 415

What is the author's opinion about the superior axon count and diameter? Why do you see the effects even for the nerve conduction velocity and not for the muscle weight?

We have added information on how the authors think about the superior axon count and diameter and the lacking effect in muscle weight. See page 16 line 334-339

Justifies the advantages of an enlarged donor site the disadvantages?

We have added information on how the authors think about the question if the advantages of a vascularized nerve graft justify the disadvantages of an enlarged donor site. Page 17 line 366-370

Can the findings of an animal study be applied to humans?

We think this is a fair question. However, we also think this cannot be said with certainty. Animal research comes closed to clinical research and is therefore the most appropriate alternative if clinical research is not available. Of course, something can be said for the fact that research in primates comes closer to humans than, say, the rat. In particular when it comes to nerve grafts because of the length and diameter of the nerve graft. Therefore, we recommended future studies to be performed under conditions more closely resembling human circumstances and to use long nerve grafts.

Therefore the Discussion section should be appropriately revised.

Minor Points

Abstract

Line 54: What does this sentence mean? Three out of four measurements. In which measurements is a vascularized nerve graft superior?

This sentence was changed to clarify it. See page 3 line 54, 55

Introduction:

The author should give a short overview of the technique of the vascularized nerve graft and explain its rationality further –

Changed, page 4 line 72-74 and 76-80 

Line 63: When is it necessary to bridge the gap – when a tensionless coaptation is not possible – this should be added

This had been added see page 3 line 64, 65 

Line 67: please rephrase the sentence – the meaning is clear – but the expression has to be improved

This sentence has been rephrased see page 3 line 66-69

Line 80: what did these studies show – please elaborate on the findings of the previous research

Information has been added see page 4 line 82, 83

Line 88: why do you use them and not nerve tubes? When you mean Axogen, then due to the results of the ranger study – these should be added.

Did you mean line 98? Why we use autografts in most cases and not allografts? Autografts give a better outcome, with the down side of donor site morbidity which is minimal for most used nerve grafts. If not enough length of autografts is present, we use axogen. 

We added references. See page 5 line 98, 99

Line 117: what about functional regeneration? Did the studies give no information focusing on the functional regeneration compared to a "conventional" ANT

No functional outcomes were described in more than one of the included articles. Only one described grasping strength and another one the sciatic function index. 

Material and Methods:

Figure 1: Can you please clarify why you exclude the 175 manuscripts to classify the arguments. Would you please adjust the figure?

Figure was adjusted accordingly.

Results:

Line 205; the meaning of the sentence is not clear – please rephrase

Rephrased see page 10 line 215

Line 210: please add the level of significance

The significance was added, see page 11 line 226

Line 235: why here SEM and not SMD?

Changed to SMD, see page 12 line 250, 251

Discussion:

This is a critical point in line 319 – 329. However, no data on functional recovery were provided, even with a small number of studies – despite the fact that this is the most important "test" for peripheral nerve injury recovery due to its clinical relevance.

We rephrased this sentence. See page 16 line 350, 351 

Line 371: Because histomorphometry is crucial to this review, the author should explain how the axonal count was obtained. Counting axons using random field of interest methods and counting the entire diameter have varying degrees of validity for the data.

We have added information on how axonal count was obtained in the Results of the manuscript. See page 11 line 220-224

---

## [Editor Report · Decision Letter 2]

18 Nov 2021

Nerve recovery from treatment with a vascularized nerve graft compared to an autologous non-vascularized nerve graft in animal models: a systematic review and meta-analysis

PONE-D-21-15417R2

Dear Dr. Broeren,

We’re pleased to inform you that your manuscript has been judged scientifically suitable for publication and will be formally accepted for publication once it meets all outstanding technical requirements.

Kind regards,

Leila Harhaus

Academic Editor

PLOS ONE

---

## [Editor Report · Acceptance letter]

22 Nov 2021

PONE-D-21-15417R2 

Nerve recovery from treatment with a vascularized nerve graft compared to an autologous non-vascularized nerve graft in animal models: a systematic review and meta-analysis 

Dear Dr. Broeren:

I'm pleased to inform you that your manuscript has been deemed suitable for publication in PLOS ONE. Congratulations! Your manuscript is now with our production department. 

Kind regards, 

on behalf of

Prof. Dr. med. Leila Harhaus 

Academic Editor

PLOS ONE